# The Relationship between the Quality of Kindergartens' Outdoor Physical Environment and Preschoolers' Social Functioning

Mariana Moreira [1,*], Rita Cordovil [1,2], Frederico Lopes [2], Brenda M. S. Da Silva [3] and Guida Veiga [4,5]

[1]  Interdisciplinary Human Performance Centre (CIPER), Faculdade de Motricidade Humana, University of Lisbon, 1499-002 Lisbon, Portugal
[2]  Laboratory of Motor Behavior, Faculdade de Motricidade Humana, University of Lisbon, 1499-002 Lisboa, Portugal
[3]  Department of Educational and Developmental Psychology, Leiden University, 2311 EZ Leiden, The Netherlands
[4]  Departamento de Desporto e Saúde, Escola de Saúde e Desenvolvimento Humano, Universidade de Évora, 7004-516 Évora, Portugal
[5]  Comprehensive Health Research Center (CHRC), Universidade de Évora, 7004-516 Évora, Portugal
*  Correspondence: marianamoreira156@gmail.com

**Abstract:** The ability to initiate and engage in relationships is a critical landmark and predictor of children's development and well-being. In kindergarten, children exhibit greater social participation outdoors rather than indoors. Indeed, the physical environment influences preschoolers' social proximity. In this study, we examine the relationship between the quality of kindergartens' outdoor physical environment and preschoolers' social functioning. Two kindergartens in Gondomar, Portugal, were selected to participate according to different levels of their physical environment outdoors (poor and fair quality) and measured by a specific physical environment rating scale. Twenty-six children (aged 3–6, 10 boys) participated in this study. Children's social proximity at the playground was measured through Radio Frequency Identification Devices (RFID). Mann–Whitney statistical tests were used to compare social proximity between groups. Our results showed that in the higher quality outdoor area, children spent less time alone and more time in social proximity with their peers in smaller groups (one or two children). More time was also spent in social proximity with different genders. Our study emphasizes the critical importance of reviewing kindergartens' outdoor physical environments to support preschoolers' social needs in a more challenging and diverse setting.

**Keywords:** kindergarten; outdoor; physical environment; preschoolers; social functioning





## 1. Introduction

Preschool children mainly interact with their peers, learn, and develop their social and emotional competencies in kindergarten [1,2]. Preschoolers are at the crucial stage of developing these interrelated competencies, impacting their daily lives and well-being in the short and long term [3,4]. Although research is scarce, kindergartens' physical environment is known to influence preschoolers' behavior and development [5].

During preschool, children expand their social bubble beyond family and develop significant relationships with peers [6]. Peers' proximity is an opportunity to socialize, learn, and practice emotional competence [7,8]. For example, when socially interacting, children must understand their own and others' emotions, which implies identifying, labeling, and discriminating emotional expressions but also associating situational contexts with those respective emotions [7]. Children who can better understand their peers' emotional expressions tend to react more prosocially and are more likable to their peers [9–11]. Theory of Mind (ToM) also has an important role in preschoolers' social success, which

is the ability to think objectively by understanding others' mental states (e.g., desires, beliefs, thoughts, intentions, and emotions). ToM enables children to explain, predict and interpret others' actions with regard to their own [12], helping them sustain social interactions. Nevertheless, preschoolers' interactions are not uniquely influenced by other children's emotional functioning. The characteristics of their surroundings, particularly the kindergarten's outdoor playground, also play a critical role [13].

In kindergarten, children exhibit greater social participation and engage in a more complex form of play with peers outdoors than indoors [2]. Outdoor kindergartens have play items (e.g., playhouse, tree, sandbox) that impose less structure on children's play, whereas indoor classrooms tend to have equipment (e.g., housekeeping area, construction area, area for painting objects) that orient children to engage in specific play activities. According to Petrakos and Howe (1996) [14], less structured equipment invites children to approach their peers, encouraging them to engage in peer interactions, particularly social pretend play.

Gibson's Affordances theory [13], which emerges from an ecological framework, can enrich our understanding of the critical role of kindergartens' outdoor physical environment in children's social interactions and relationships [15]. According to this theory, affordance relates to the animal's (in this case, the child's) perception of environmental features. Specifically, affordances in the kindergarten's outdoor physical environment refer to what is provided, perceived, or recognized by children as realizable, relating to their needs, interests, motivation, or capabilities [16]. Thus, the physical environment enables or hinders children's social proximity to their peers, orients children's experience, and conveys psychological meanings [17] depending on whether it facilitates or inhibits their exploration [16,18].

Hence, creating outdoor environments with a wide variety of affordances and resources ensures that play and exploration are of paramount importance [19,20]. Educators have a decisive role in creating such contexts and opportunities, enabling children to engage in different play behaviors and social proximity that promote their development and learning [21,22].

Previous studies have explored the relationship between certain elements of outdoor kindergartens with children's peer play behaviors [2,15,17,23,24]. For example, providing enough diversity, surface variety, and play materials promote toddlers' exploratory behaviors, engagement, and social interactions [25]. Different and well-defined play yards (e.g., space for social and fantasy play) increase preschoolers' interactions and cooperative behaviors [26]. Small outdoor spaces invite preschoolers' physical togetherness, whereas larger ones invite them to engage in larger group activities (e.g., team sports) [27]. Secret or retreat places are also associated with increased peer play [28]. Natural elements such as plants, water, and the presence of vehicles (e.g., bikes) [29] are associated with increased peer relationships and decreased antisocial behavior [23]. Focusing on these features, Moore [30] developed an index to assess the quality of outdoor physical environments. According to Moore, an outdoor physical environment was good if it had diverse surfaces, large and small playing areas, social and fantasy play spaces, a friendly climate, safe yet challenging contours, and secret or retreat places [30].

Previous studies focusing on preschoolers (aged 3–6 years) have examined the relationship between particular characteristics of the kindergarten's outdoor physical environment and children's social play and behavior [15,31]. These studies have shown that preschoolers are more likely to engage in the most complex form of peer play (i.e., interactive dramatic play) outdoors than indoors [2]. The more opportunities the playground offers the child (variety of surfaces and materials), the more the child tends to play close to peers or in groups [15]. In this outdoor environment, children spend less time playing alone and more time cooperating with their peers in small or large groups [31]. In these studies, the children's social behavior was coded through play observation scales using systematic observation methods, which can sometimes be intrusive and subjective [32]. Only one of these studies [2] performed a qualitative characterization of kindergartens' outdoor phys-

ical environment and physical attributes (variety and complexity of available materials). This approach weakens a more objective and consistent characterization of kindergartens' outdoor physical environment. Furthermore, none of these studies objectively analyzed the relationship between the kindergarten's physical environment outdoors and children's social functioning (e.g., proximity to peers).

Thus, to our knowledge, ours is the first study using quantitative and objective instruments to examine the relationship between kindergartens' outdoor physical environment and the preschoolers' social functioning.

The preschool years are a critical transition stage when children develop the fundamental motor and social-emotional skills that enable them to "collaboratively self-regulate their cooperative interactions with others" [33] (p. 9). In preschool, children engage in social play, cooperate to attain a common goal, and engage in longer interactions with their peers [34]. In addition, preschoolers show a natural tendency to socialize with same-gender partners [35], especially in kindergarten [36]. Previous studies suggested that such a tendency to interact with same-gendered peers is associated with positive emotions for young boys and girls, which is an easier way for children to feel accepted by their same-gendered peers and find someone with the same play interests [36]. However, other studies show that restricting the social network to same-gender peers negatively impacts children's short- and long-term development [36–38].

Thus, we aimed to compare the social functioning of preschoolers attending a kindergarten with a higher outdoor quality versus those attending a kindergarten with poorer outdoor quality. Emotional competence, considering its association with social functioning, was controlled in this study. In line with previous research [2,15,31], we hypothesized that children attending a kindergarten with a higher-quality outdoor physical environment spend more time in social proximity with their same- and different-gender peers and less time alone. We hypothesized that the opposite would happen to children attending playgrounds with a poorer-quality physical environment. Understanding these differences will provide insights into which kindergarten's outdoor physical environment characteristics promote preschoolers' social proximity. Such knowledge is more important than ever, considering the negative consequences of the COVID-19 lockdown on children's social experiences [1].

## 2. Materials and Methods

### 2.1. Kindergartens' Quality Assessment and Selection

Before data collection began, an initial assessment of the kindergarten's physical environment quality was conducted for 18 kindergartens in Gondomar (the Porto area of Portugal). The kindergarten's outdoor physical environment quality was assessed with the *Escala de Avaliação dos Envolvimentos Físicos para Crianças* (EAEFC) (Children's Physical Environments Rating Scale) [39]. This scale is the Portuguese version of the Children's Physical Environments Rating Scale (CPERS5) [40]. The EAEFC aims to holistically characterize the quality of the kindergarten's physical environment and analyzes whether or not the physical features of the indoor and outdoor spaces comply with the indicators promoting favorable childhood development. This scale is divided into 124 items spread through 14 subscales and is grouped into 4 parts (for more details, see Moreira et al., 2020 [39]). The subscale's items are scored from 0 (Do not comply) to 4 (Comply with excellence). A final subscale score is obtained by the average values of all the items. In turn, it is possible to obtain a final quality score for the kindergarten's physical environment by calculating the average values of all the subscales. Both scores range from 0.00–1.00 = bad quality; 1.01–2.00 = fair quality; 2.01–3.00 = good quality, and 3.01–4.00 = excellent quality [40]. Depending on the study's goals, the subscales may be applied and quoted independently, providing information about the quality of a kindergarten's physical environment. Only the EAEFC 13 subscale (8 items) named Espaços Exteriores de Jogo: Necessidades de desenvolvimento (Play Yards-Developmental Needs)was considered for this study. The

characteristics of the outdoor space, surface variety, areas and elements, and the balance between safety and risk were assessed for this subscale in each kindergarten (see Table 1).

**Table 1.** Description of the items for the *Escala de Avaliação dos Envolvimentos Físicos para Crianças* (EAEFC) 13 subscale, Espaços Exteriores de Jogo: Necessidades de desenvolvimento (Play Yards-Developmental Needs).

| Subscale | Items Examples |
|---|---|
| 13. Play Yards–Developmental Needs | 13.1 The play yards provide enough diversity, such as various surfaces for different types of play, to interest children (e.g., grass, hard surfaces, sand). |
| | 13.2 The play yards have both large and small areas for children to play. |
| | 13.3 The play yards have space for social and fantasy play (e.g., quiet areas away from physical play, cubby house, outdoor playhouse, storage for dress-up props, etc.). |
| | 13.4 Some of the play yards are smaller and have a friendly feeling (e.g., intimate character, natural elements, etc.) |
| | 13.5 Some of the play yards have contours that are safe yet challenging enough for children to play on. |
| | 13.6 Secret or retreat places exist for a child to be alone yet within sight of adults. |
| | 13.7 There is a garden that children help maintain (ask the director if necessary). |
| | 13.8 There is an identifiable area for outdoor water play (e.g., outdoor water table, tap, sprinklers, natural ponds, etc.). |

Note: For more information on the scale items, see Moore et al., (2003). Reprinted with permission from [41]. 2003, Children's Physical Environments Rating Scale.

A researcher completed the scale via direct observation of the kindergarten's outdoor physical environment and developed on-site during visits to each preschool lasting from 45 to 50 min. The scores on the EAEFC 13 subscale, Play Yards- Developmental Needs, ranged from bad to good quality (M = 0.94; DP = 0.77; min. = 0.00; max. = 2.50). After the initial assessment, two kindergartens were selected to participate based on higher and lower scores. Kindergarten A was one of the institutions with the lowest final score (0.00), and Kindergarten B's quality was fair (1.75). The kindergartens with higher scores were contacted but did not accept the invitation to participate.

Kindergarten A was from the public sector and hosted only one preschool class with 19 children. The outdoor playground area was 25 m$^2$ (1.67 m$^2$ per child). The floor was cement with just one floor marking with a hopscotch game and no fixed equipment. Some loose materials were available, although only for symbolic play (e.g., dolls and puppets, cars). Part of the playground was covered (see Figure 1).

Kindergarten B was from the private sector. It had 66 children, ranging from crawlers, walkers, and preschoolers, to first-grade children. In this preschool, only the facilities for crawlers, walkers, and preschoolers were assessed by the EAEFC scale. The outdoor l area was 87.50 m$^2$ (7.95 m$^2$ per child). The outdoor environment offered large grass spaces and small cement spaces with distinguishable play areas (i.e., a big sandbox, grassy area, fixed swing, small bench, and undefined areas). There were some trees that children could climb. There was no roof protection in this outdoor space. Some loose materials (plastic cups and shovels), small toys, and box games (e.g., puzzles, bicycles, and a portable slide) were also accessible to children (see Figure 2).

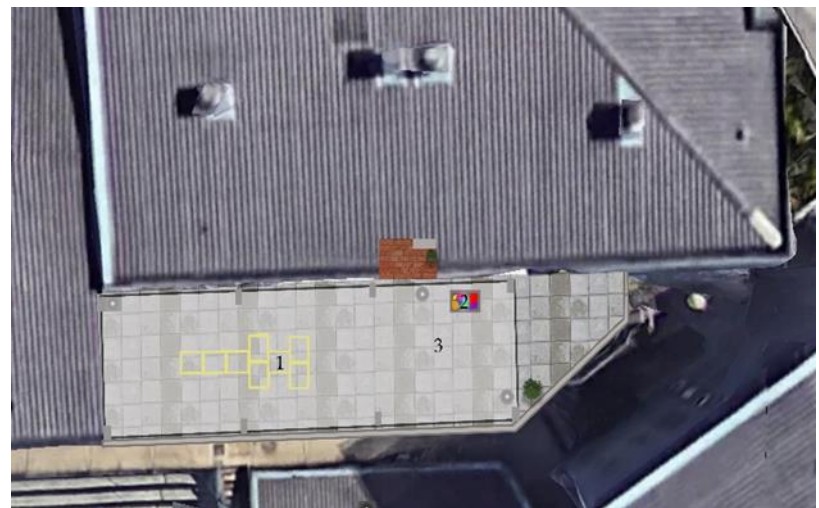

**Figure 1.** Kindergarten A's outdoor physical environment: (1) hopscotch mark floor; (2) box with symbolic play materials; (3) part of the outdoors, which is roofed.

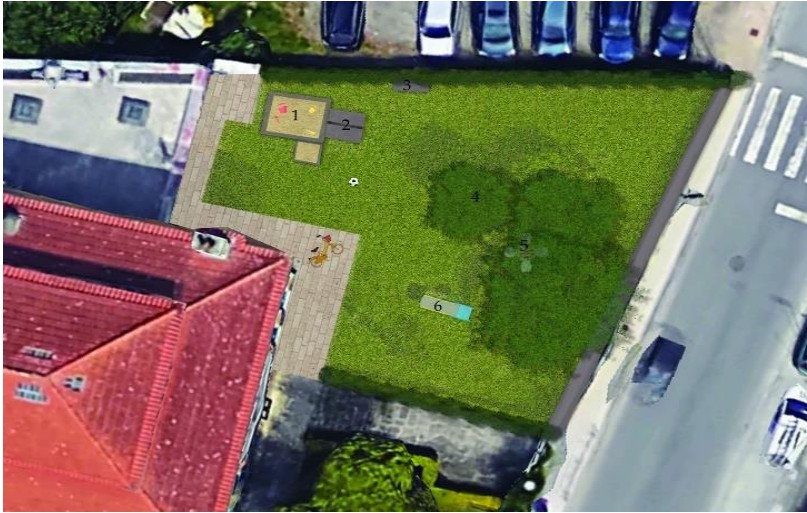

**Figure 2.** Kindergarten B's outdoor physical environment: (1) sand box with loose parts; (2) vintage water pump that is not working; (3) small fixed bench; (4) tree to climb; (5) fixed swing; (6) portable slide.

*2.2. Participants*

Twenty-six children with typical development (10 boys and 16 girls) between 3- to 6-years-old (M = 4.80 ± 0.78 years) participated in this study. One class from each kindergarten was chosen to participate in the study based on their educators' availability. The class from Kindergarten A had 15 preschool children (7 boys and 8 girls; M = 4.47 ± 0.63 years), and the class from Kindergarten B had 11 (3 boys and 8 girls; M = 5.33 ± 0.71 years). Families in Kindergarten A were mainly of middle socioeconomic status (80%) and upper socioeconomic status in Kindergarten B (55.6%).

This research was conducted according to all respondents' human rights and capabilities while acknowledging and aligning with the Ethical Code for Early Childhood Researchers (EECERA) [42]. Preschools and caregivers were informed about the goals and procedures of the study, how data would be analyzed and saved to guarantee their privacy, and the voluntary nature of their participation. Caregivers provided their written consent, and children gave verbal consent before data collection. They were also informed that they could quit at any moment without consequences. The researcher (with educators'

collaboration) prepared the first meeting with children in their activity room to ensure they received the same information. Children were informed about the study's aims in a simple and ludic way. It was explained to children that they were not obligated to go outdoors if they did not want to. In both kindergartens, children's routines were respected. To ensure data confidentiality, a numerical code was assigned to each participant (known only to the researcher). The researcher has committed to returning the study results to the kindergartens.

### 2.3. Procedures

Data collection in the two kindergartens transpired during the pandemic from January 2021 to April 2021. After obtaining parental consent and children's verbal assent to participate, parents filled out questionnaires about socioeconomic and demographic information.

Children's emotional competence was tested individually in sessions that took approximately 20 min.

Children's peer social proximity to outdoor environments was measured using proximity-sensing Radio Frequency Identification Devices (RFID) badges during one recess of free play time (30 min). Due to COVID-19 restrictions, only two days for data collection with RFID sensors in the kindergartens were possible. However, only the data from one of the collection days were used. The first day was dedicated to acclimating children to the badges.

The researcher was in each preschool one week before data collection, so children grew accustomed to her presence, the cameras, and badges. During this time, the researcher also conducted the necessary pretests to ensure a reliable RFID application. After the first week, children ignored the cameras and researcher and also forgot the badges during playful social proximity (for more details, see Veiga et al., 2017 [43] and Elmer et al., 2019 [44]), which allowed data collection with the RFID sensors to start. On the day of data collection, RFID badges (3 × 3 cm plastic squares) were attached to the children's clothes. The signal strength between routers and badges and the batteries were tested for each sensor before starting data collection.

All procedures were conducted following the 1964 Helsinki declaration and its later amendments. The Ethics Committee approved this study.

### 2.4. Instruments

2.4.1. Peer Proximity

Children's social proximity was assessed using proximity RFID sensors attached to the clothes of all participating children. When children were within a radius of approximately 1.5 m and facing each other, a social contact was registered by the receiving stations (i.e., Ethernet reader) [44,45]. The RFID system measures all mutual and multiple contacts at a sampling rate of 4 Hz (4 measurements per second). To improve measurement quality and overcome possible measurement biases, an interpolation method was used with a time threshold of 20 s. In this sense, contacts that lasted <20 s were not registered as social proximity, and ongoing proximity interrupted for <20 s was considered a single social proximity [44]. Ongoing proximity interrupted by >20 s counted as separate social proximity [44]. The use of this correction was important for two main reasons: Firstly, in a free play time context, children often physically move away even though they are still involved in social proximity (e.g., a game of catch). Secondly, when children are in social proximity with others, different factors (e.g., the way they are moving, or the presence of other objects or children), may interrupt the signal.

RFID sensors allowed us to extract multiple variables, such as the number, duration and type of social proximity. Periods in which badges did not detect any proximity were defined as alone time. Different variables per child were thus obtained.

- *Percentage of time in proximity with peers*: Total time child spent interacting with other children (in seconds), divided by the time the child's badge was detected;

- *Percentage of time in proximity in dyadic*: Total interaction time with only one child (in seconds) divided by the time the child's badge was detected;
- *Percentage of time in proximity with two children:* Total interaction time with two children (in seconds) divided by the time the child's badge was detected;
- *Percentage of time in proximity with three or more children*: Total interaction time with three or more children (in seconds) divided by the time the child's badge was detected;
- *Percentage of time in proximity with same gender:* Time interacting with same-gender peers (in seconds) divided by the time spent with the same and opposite genders;
- *Percentage of time in proximity with different gender:* Time interacting with the opposite gender divided by the time spent with the same and opposite genders;
- *Percentage of time alone:* Total time child spent alone (in seconds) divided by the time the child's badge was detected.

To conduct the proximity analysis by gender, the number associated with each child's name was associated with a code for the child's gender.

### 2.4.2. Emotional Competence

Two components of emotional competence were measured: *Emotion Understanding* and *Theory of Mind* (ToM) [46,47].

*Emotion Understanding* comprised two tasks for emotion recognition (discrimination and identification of facial expressions) and one for emotion attribution. A composite score was computed based on these tasks, as in a previous study [45].

ToM was evaluated with a Desire Task and two False Belief Tasks. For each task, children received a maximum score of 2 points. A composite score was computed based on these tasks, as in a previous study [47,48].

### 2.5. Data Analysis

A descriptive analysis was performed for all variables. Due to the non-normality of the data, Mann–Whitney statistical tests were used to investigate differences in the emotional competence and social functioning variables.

## 3. Results

Our results showed that children's emotional competence did not differ between the two kindergarten groups (see Table 2).

**Table 2.** Descriptive statistics (Mean and SD) and Mann–Whitney test results of emotional competence variables.

| Emotional Competence | Range | Kindergarten | | Mann-Whitney |
|---|---|---|---|---|
| | | A (*n* = 15) M (SD) | B (*n* = 11) M (SD) | *p* |
| Emotion understanding | −3–3 | −0.06 (0.73) | 0.57 (0.64) | 0.178 |
| Theory of Mind (ToM) | 1–2 | 0.58 (0.21) | 0.60 (0.30) | 0.422 |

Table 3 shows differences between the social functioning variables in the two kindergartens, including time with peers, time alone, time with two children, time in proximity, in dyadic, and time in social proximity with the opposite gender.

Children from Kindergarten B (with a higher-quality playground, i.e., with various physical opportunities) spent less time alone and more time in proximity with their peers in smaller groups (with 1 and 2 children), and more time in proximity with the opposite genders compared to the children from Kindergarten A (less-quality playground) (see Table 3).

**Table 3.** Descriptive statistics (Mean and SD) and Mann–Whitney test results of social interaction variables in the two kindergartens.

| Social Variable | Kindergarten | | Mann-Whitney |
|---|---|---|---|
| | A (n = 15) | B (n = 11) | |
| | **M (DP)** | **M (DP)** | ***p*** |
| % time alone | 41.75 (0.06) | 6.91 (0.07) | 0.000 * |
| % time with peers | 58.25 (0.06) | 93.09 (0.07) | 0.000 * |
| 　% time $\geq$ 3 children | 57.00 (0.07) | 47.27 (0.22) | 0.443 |
| 　% time with 2 children | 0.16 (0.00) | 23.18 (0.06) | 0.000 * |
| 　% time in dyadic proximity | 0.96 (0.03) | 3.82 (0.19) | 0.001 * |
| % time with = gender | 0.91(0.08) | 0.77 (0.23) | 0.148 |
| % time with $\neq$ gender | 0.09 (0.08) | 0.28 (0.23) | 0.009 * |

Notes: * significant differences for $p < 0.05$; Kindergarten A—poor play yard—25 m$^2$; Kindergarten B—fair play yard—87.5 m$^2$.

## 4. Discussion

Kindergartens' outdoor playgrounds are one of the major contexts where preschoolers interact with their peers [2,49]. After the COVID-19 lockdown period, which limited children's social interactions [1], outdoor playgrounds became even more fundamental for children to fulfill their social needs.

In line with our hypotheses, outdoor physical quality was associated with outdoor social functioning. In a better physical environment, children spent more time with their peers and less time alone than in an outdoor with a poor physical environment, suggesting that a more child-friendly outdoor physical environment is associated with social closeness and more opportunities for interactions to happen [2,15,24]. Playgrounds with various and diverse surfaces (e.g., grassy areas, asphalt), natural play materials, functional equipment (e.g., slides, fixed swing), and loose parts (e.g., balls, sand toys) are important for social proximity. A greater variety and diversity of play surfaces and materials may meet the interests of many children and encourage engagement. In this way, children may feel more comfortable playing and interacting with peers for extended periods. Additionally, the availability of specific types of play materials (e.g., fixed swings, sandboxes, trees to climb, sand toys) may promote children's engagement in physical play [50,51] and loose parts play [23], which are both associated with preschoolers' social functioning [23,43]. Moreover, children spent less time alone in a higher-quality play yard with defined and different play areas. Different materials, toys, and floor surfaces divided into different play areas may broaden possibilities for social proximity, as previously argued [13,18,52]. Interestingly, in another study involving the outdoors and preschool children, an increased number of nearby play settings and moveable loose parts led children to be more active and mobile between settings to join peers in other play and social activities [50]. In other words, a diversified range of behavior settings contributes to a more challenging play environment, facilitating group and collaborative engagement.

Furthermore, a higher-quality outdoor physical environment promotes more mixed-gender social proximity, a potential opportunity to enrich critical social skills [36]. Our findings suggest that having an outdoor playground with more natural elements (such as sand or trees) and gender-neutral toys and play areas (see-saw, slide, bench) may promote more opportunities for contact between mixed-gender peers. On the contrary, the absence of play areas and the presence of small, symbolic, more gender-stereotyped toys (e.g., cars, dolls) may prevent contact between mixed-gender peer groups. It is important to note that the preference for certain toys/equipment is conditioned by stereotypes and social expectations regarding gender-specific behavior. Adult culture also affects and mediates children's own interpretative culture and behaviors [53]. For example, during pretend and other types of play, children are known to enact experiences of personal, interpersonal, domestic, social, and cultural situations, where elements of identity, gender, culture, and

ethnicity, among others, are embedded [54]. Nevertheless, an outdoor physical environment with well-defined micro-settings may guarantee the playground's necessary structural, organizational, and emotional safety, encouraging children to play with opposite-gender peers and creating moments to experience the benefits of mixed-gender interactions.

The higher-quality outdoor physical environment also promoted social proximity in smaller groups, possibly suggesting more intimate relationships. In this kindergarten, there were benches and tree shade, which, according to Heft (1988) [55], are physical features that provide a space to sit, privacy, and refuge. Our findings align with previous studies on children's social behavior in indoor environments. These studies showed that circumscribed zones (or semi-open arrangements) foster peer affiliative interactions [56,57], and promote and sustain younger children's interactions with their peers [22]. The opposite occurs in open arrangement (e.g., no circumscribed zones available) and close arrangement settings (e.g., where physical boundaries divide the locale in two, obstructing children's view of the entire space) [56,57], which promote fewer peer interactions. In an open arrangement, children are dispersed and frequently move from place to place, whereas in a closed arrangement, children tend to avoid areas where the adult is out of view and stands close to them [56,57]. Furthermore, the unstructured context (e.g., a large empty area with scarce furniture, equipment, or objects), does not facilitate peer interactions [56,57]. Moreover, previous research showed that the variety and diversity of play surfaces, materials, and play areas help children engage deeply in peer relationships [26,57].

There is still no consensus regarding the adaptive nature of being alone on the playground at preschool age [45]. While some children choose to be alone to have time for reflection and quietness [58], others are alone because they cannot engage in or maintain interactions with their peer group or are excluded [59]. Some studies have shown that playing alone may also predict poor academic performance, poor social functioning, and mental health problems in later childhood and adulthood [60,61]. Nevertheless, our findings suggest the importance of the quality of the physical environment in preventing loneliness in children. In particular, increasing the variety of surfaces, play materials, and play areas may be necessary for children's social needs. Considering preschoolers are still developing their language skills [62], the possibility of engaging in less language-dependent types of play and move independently and actively between different outdoor play areas may help them initiate and sustain interactions with their peers and spend less time playing alone [63].

Our findings on children's social proximity suggest that higher-quality outdoor environments with natural features promote more proximity and mixed-gender contact between children. These results align with a theoretical model proposed in a recent systematic review on the associations between nature-based early childhood education and children's social, emotional, and cognitive development [64]. In this work, nature-based kindergartens offer the possibility for children to increase their creativity and imagination, diversify play opportunities, and instigate prosocial interactions among peers and educators, which play a key role in improving social skills.

*Strengths, Limitations, and Future Directions*

Our study is the first to examine the relationship between the quality of kindergartens' outdoor physical environment and preschoolers' social functioning. We used RFID sensors and a quantitative scale to objectively measure social functioning and assess the quality of kindergartens' outdoor physical environment, respectively.

In future studies, it would be interesting to combine this scale with instruments focused on the social opportunities provided by each playground element, such as Kyttä's (2002) functional taxonomy [16] based on Heft's (1988) affordances taxonomy of children's outdoor environments [55] but considering sociality. Furthermore, it is also important to know children's perceptions about the kind of outdoor elements that foster their interactions with peers. Participatory methodologies, such as interviews and drawings, could be used to collect children's perceptions. Other studies have used children's participa-

tory methodologies in redesigning outdoor spaces by listening to their perspectives on meaningful play features [65–67].

Due to COVID-19 constraints, only two kindergartens chose to participate in this study, which limited the initial assessment. Involving more kindergartens increases the likelihood of comparing those with even more marked differences in the quality of outdoor physical environments.

Using RFID sensors to measure social functioning is a valuable strategy, overcoming the well-known limitations of questionnaires or observations [32] and capturing intense and continuous data about children's social proximity [68–70]. However, as previously noted, these sensors only measure face-to-face proximity and miss the contact quality. Hence, future studies should examine peer interactions beyond social proximity, for example, by combining these data with other observational methods to gain deeper insight into children's social relationships. We also encourage future studies to collect data with RFID sensors for more days to better generalize the data.

We also suggest that future studies consider parents' and educators' perceptions about outdoor physical environments in kindergartens and how they contribute to children's social development [71,72].

To our knowledge, the present study provides an innovative view of the quality of outdoor physical environments in kindergartens and children's social functioning measured by social proximity. Moreover, it is important to design outdoor kindergarten environments considering preschoolers' social behavior.

## 5. Conclusions

The outcomes of this study show that a higher-quality outdoor kindergarten fosters preschoolers' ability to initiate and sustain social proximity to peers and leads to less solitary behavior and more time in small and mixed groups. A noteworthy contribution to this study was measuring the quality of the physical environment as an exclusive variable and examining children's social functioning through an intensive, continuous, and non-obtrusive method (RFID sensors).

These results reinforce that the children's proximal environment becomes more socially meaningful if its design promotes mobility, physical activity, diversified affordances, and contact with natural elements. Such interactive and multi-dimensional outdoor micro-systems positively impact children's social and adaptive behaviors.

In addition, the present study's findings and innovative methodology reinforce the need for educational and political agents to review kindergartens' outdoor physical environments and foster children's social experiences. After the lockdown period, children's relationships with peers and emotional well-being were compromised [1]. Therefore, it is crucial that researchers, educators, policymakers, and children work together to ensure that the outdoor kindergarten environment remains a dynamic physical and social place that supports children's play, well-being, development, and educational thriving.

**Author Contributions:** Conceptualization, M.M., R.C., F.L. and G.V.; Methodology, M.M., R.C., F.L. and G.V.; Software, B.M.S.D.S.; Validation, M.M. and G.V.; Formal Analysis, B.S and G.V.; Investigation, M.M.; Resources, M.M., R.C., F.L., B.M.S.D.S. and G.V.; Data Curation, M.M and G.V.; Writing—Original Draft Preparation, M.M. and G.V.; Writing—Review & Editing, M.M., R.C., F.L., B.S and G.V.; Visualization, M.M. and G.V.; Supervision, R.C., F.L. and G.V.; Project Administration, R.C., F.L. and G.V.; Funding Acquisition, M.M., R.C. and G.V. All authors have read and agreed to the published version of the manuscript.

**Funding:** This research was partly supported by the Portuguese Foundation for Science and Technology, under Grant UIDB/00447/2020 to CIPER—Centro Interdisciplinar para o Estudo da Performance Humana (unit 447). Mariana Moreira was supported by the Fundação para a Ciência e Tecnologia (FCT), Portugal (grant number SFRH/BD/138071/2018).

**Institutional Review Board Statement:** The research was conducted in accordance with the Declaration of Helsinki and its later amendments, and approved by the Ethics Committee of the Faculty of Human Kinetics, University of Lisbon (CEIF Approval Number: 26/2019).

**Informed Consent Statement:** Written informed consent was obtained from schools and parents to publish this paper.

**Data Availability Statement:** The data presented in this study are available on request from the corresponding author. The data are not publicly available due to privacy or ethical restrictions.

**Acknowledgments:** We thank the kindergartens for the warmth with which they received this study, the educators for their availability and motivation, and the parents and children for agreeing to participate in the study, making it possible to happen. We also want to thank Joana Jacinto for helping us to draw the sketches of the outdoors of each kindergarten and for her contribution to the graphical abstract so that they could illustrate and enrich this article. We also thank Carolien Rieffe and Maedeh Nasri from Focus on Emotions Group (Leiden University) for the RFID sensors and the IT support. Finally, we thank the research group Development and Education, from the Center for Psychology at the University of Porto (CPUP), for the cameras.

**Conflicts of Interest:** The authors declare no conflict of interest.

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
