# Peer review of "The Relationship between the Quality of Kindergartens’ Outdoor Physical Environment and Preschoolers’ Social Functioning"

_education, doi:10.3390/educsci12100661_

Round 1

Reviewer 1 Report

Thank you for submitting this paper.  I do hope the following recommendations will support you in improving it to a sufficiently high standard for publication.

1. In the Introduction section please state what age range of children you are referring to, or the literature you cite refers to. An assumption seems to be made that babies, toddlers and preschoolers interact with the outdoor environment in the same way, which is not necessary true.

2. It would would helpful to clarify what are the particular features of outdoors and outdoor play are distinct from the indoor environment and play in an indoor environment.

3. There is no reference at all to the role of the educator with respect to children's social functionning.  This is a serious omission and needs addressing.

4. Recommend you consider the empirical work of Maria Clotilde Rosetti-Ferreira and colleagues from Brazil undertaken over many years, which consider influence of arrangement of space on peer relations of infants and toddlers. This work is published both in English and Portuguese and so should be accessible to you.

5. The discussion on gender segregation (line 88-90; 308-316) needs a much more nuanced analysis, discussion and conclusion.  Just because children choose to select same-gender play partners, does not mean this is forced segregation. 

6. The use of RFID badges is an interesting and unusual data collection tool. However, basing findings on just one 30 minute observation during one recess time in just two settings is a serious limitation which weakens the study considerably.   Please explain why repeated measures over a number of days using the RFID badges were not carried out.  Given this limitation findings and conclusions should be more tentatively expressed. This issue also needs to be acknowledged in the limitations section. 

7. Clarify why more recently published ethical guidelines in educational research were not followed, such as guidelines of BERA or EECERA. 

8. A few small typos/spelling errors identified: line 67, focussing; line 109, holistically.

Wishing you well as you review your paper.

Reviewer 2 Report

The subject of study is of great interest.

Perhaps the resume should indicate the specific place and the type of analysis carried out, briefly

The subject of study is of great interest.

Perhaps the resume should briefly indicate the specific place and the type of analysis carried out.

The method is fine.

The procedure is fine.

The job structure is fine.

Studies like the one presented are needed.

I ask the authors to consult the works and discuss the data of this study with this other work

ASSESSMENT AND SATISFACTION OF FAMILIES WITH OUTDOOR MOTOR LEARNING SPACES IN ALBACETE, SPAIN Pedro Gil-Madrona and María Martínez-López. Interscience. December 2016. Vol 41, nº 12. 833-837

And with this other job

Gil-Madrona, P.; Martinez-Lopez, M.; Prieto-Ayuso, A.; Saraiva, L.; Vecina-Cifuentes, J.; Vicente Ballesteros, T.; Moratilla-López, R.; López-Sánchez, GF Contribution of public playgrounds to motor, social and creative development and reduction of childhood obesity. Sustainability 2019, 11, 3787. https://doi.org/10.3390/su11143787

0378-1844/14/07/468-08

Congratulations to the authors

Reviewer 3 Report

The study reviewed is interesting as it relates to the social functioning of preschoolers with the quality of the kindergarten outdoor physical environment. The social interactions of 26 children (3 to 6 years old) attending two kindergartens with different quality levels of their outdoor physical environment (one poor, the other fair) were studied and measured using Radio Frequency Identification Devices. The results recorded higher levels of social interactions in the children of the kindergarten with the richest outdoor physical environment.

I am not entirely convinced of two serious issues about this research. In particular:

1)      I do not understand the originality and necessity of this research as it seems that everything that is being studied is already known from the literature. It would help if the authors would give some more arguments to support the necessity and especially the originality of their study.

2)      I do not understand how the social interactions of infants are recorded. From the descriptions in the methodology, I understand that what is recorded is if the signals emitted by the children appear to be approaching 1.5 meters. But then it is not at all obvious if there is interaction, either verbal or other types. How do they come to that conclusion? Also, how is the gender of the children recorded, and then there are results for interactions between children of the same or different gender? How do they know what the gender of the child in a particular emission is? If we don't have additional descriptions, details, and documentation for me there is a serious question of the validity of the whole study.

Some other issues:

3)      In the abstract, the sentence "A Kindergarten’s outdoor physical environment positively influences children's social interactions by offering various play surfaces and materials, 6 well-defined play areas, and a challenging design." (Page 1, lines 5-7) is worded in such a way that one wonders if this is known with such certainty, why this research?

4)      About the picture on page 1. I think firstly it needs a caption. Secondly, it's a pictorial summary of the whole study, very cute, if the authors want it to be there any way it might be better to move it to the end of the results section.

5)      On page 2, lines 30-32, I sense that the terms “competencies” and “skills” are used as synonyms when they are not. It should be reworded.

6)      On page 1, lines 71-72: “Also, in a natural outdoor playground environments …”. A rewording is needed.

Round 2

Reviewer 1 Report

Attention paid to revising paper based on earlier review is noted. I recommend publishing in current revised form.

Author Response

We thank the reviewer for all the care and rigor he has reviewed our article, allowing us to improve its quality.

Reviewer 3 Report

The authors more or less responded adequately to the comments and remarks I made in the previous round of the review process, except for one point. This was my questioning of whether the methodology used recorded social activity of infants, or simply that two or more children, were less than 1.5 meters away, facing each other, for a continuous time of 20 sec. Since the authors honestly state that they did not utilize the camera to record whether social interactions were actually occurring, I think this research has serious methodological limitations that should be documented throughout the paper. That is, I believe that this is about potential social behaviour or opportunities for social interaction between infants and so should be reported throughout the text.
